# Influence of a Flat Polyimide Inlay on the Propagation of Guided Ultrasonic Waves in a Narrow GFRP-Specimen

**DOI:** 10.3390/ma15196752

**Published:** 2022-09-29

**Authors:** Liv Rittmeier, Thomas Roloff, Natalie Rauter, Andrey Mikhaylenko, Jan Niklas Haus, Rolf Lammering, Andreas Dietzel, Michael Sinapius

**Affiliations:** 1Institute of Mechanics and Adaptronics, Technische Universität Braunschweig, Langer Kamp 6, 38106 Braunschweig, Germany; 2Institute of Mechanics, Helmut-Schmidt-University/University of the Federal Armed Forces Hamburg, Holstenhofweg 85, 22043 Hamburg, Germany; 3Institute of Microtechnology, Technische Universität Braunschweig, Alte Salzdahlumer Str. 203, 38124 Braunschweig, Germany

**Keywords:** structural health monitoring, narrow specimen, guided ultrasonic waves, continuous wavelet transformation, numerical simulation, composite materials, GFRP

## Abstract

Structural health monitoring systems for composite laminates using guided ultrasonic waves become more versatile with the structural integration of sensors. However, the data generated within these sensors have to be transmitted from the laminate to the outside, where polyimide-based printed circuit boards play a major role. This study investigates, to what extent integrated polyimide inlays with applied sensor bodies influence the guided ultrasonic wave propagation in glass fiber-reinforced polymer specimens. For reasons of resource efficiency, narrow specimens are used. Numerical simulations of a damping-free specimen indicate reflections of the $S_{0}$-mode at the integrated inlay. This is validated experimentally with an air-coupled ultrasonic technique and a 3D laser Doppler vibrometry measurement. The experimental data are evaluated with a method including temporal and spatial continuous wavelet transformations to clearly identify periodically occurring wave packages as edge reflections and distinguish them from possible inlay reflections. However, even when separating in-plane and out-of-plane movements using the 3D measurement, no reflections at the inlays are detected. This leads to the conclusion that polyimide inlays are well suited as substrates for printed circuit boards integrated into fiber-reinforced polymer structures for structural health monitoring, since they do not significantly influence the wave propagation.

## 1. Introduction

Structural health monitoring (SHM) using guided ultrasonic waves (GUW) has been of great interest in research for the detection of structural damage [1,2,3,4]. GUW can propagate over long distances in thin-walled structures without dissipating much energy and interact with acoustic impedance changes caused by inhomogeneities such as damage [4]. These interactions result in reflections, scattering, mode conversions and amplitude attenuation, which can be observed in the wave field [5,6]. Therefore, changes in the wave propagation are an indicator for damage, which can be detected using different measurement techniques such as laser vibrometry [7], air-coupled technique [8] or integrated piezoelectric sensors [1]. Air-coupled ultrasonic technique and laser vibrometry allow inspections of the wave field at the surface of the structure only at specific inspection intervals. In contrast to this, applied, discrete sensors enable permanent monitoring of the structural health even during operation.

To obtain information about the wave propagation within the structure, it is useful to integrate sensors at the location of the event. In a previous work, the authors presented a novel embeddable microelectromechanical system (MEMS) vibrometer made mainly out of borosilicate glass for GUW detection [9]. However, the integration of sensors causes a local change in stiffness, which leads to a weakening of the structure and a local change in acoustic impedance. Additionally, electrical contacts such as state-of-the-art polyimide-based printed circuits boards (PCB), are equally needed and have to be integrated into the laminated structures. This can affect the propagating wave field and the integrated sensor acts as a local defect. Nevertheless, separating the interference caused by sensors or damage is crucial for a reliable SHM system.

The characteristics of wave field interferences with damage are extensively investigated in the literature. Typical types of damage include cracks, corrosion, impact, delaminations and through-holes. Within the classical approach, which is usually referred to as linear wave propagation, the detection of damage smaller than the wavelength is not possible [10]. Due to this limitation, a second approach has been profoundly investigated in recent decades, which deals with nonlinear wave propagation. This approach is based on the generation of higher harmonic modes due to microstructural damage [11,12,13,14]. This effect is detectable within the linear wave propagation by a loss of energy recognizable in wavelet decomposition plots [15]. The work presented herein focuses on the development and integration of sensors to detect or observe the wave propagation in fiber metal laminates. Critical damage scenarios in this class of materials are severe delaminations due to low-velocity impact [16,17]. Since these delaminations are typically at least within the size range of the considered wavelength, only linear wave propagation is analyzed here. The influence of delaminations and impact damage on linear wave propagation has been intensely investigated over the years [5,6,15,18,19,20,21]. It is shown that the scattered and converted mode amplitudes depend on the fiber orientation, the delamination size as well as the relative position along the laminate’s height [6]. The amplitude increases with the delamination size [6,15,22], the time-of-flight is affected [15] and mode conversion occurs [5]. In addition, the wave field is also significantly influenced by through-holes [15] causing wave scattering [23]. Moreover, the interaction of GUW with material discontinuities is investigated experimentally and numerically regarding welded joints of dissimilar materials [21]. The wave interaction is described by defining transmission and reflection coefficients. Mode conversion is observed; however, it should be noted that the mode conversion is not only induced by damage, but is also strongly related to the random scattering of the material parameters in fiber-reinforced composites (FRP) [24,25].

As shown, there is extensive work on the interference between damage and GUW whereas the influence of integrated sensors is hardly addressed. In contrast to discrete sensors, the integration of optical fiber sensors [26] and fiber Bragg grating (FBG) sensors [27,28,29] is extensively investigated in the literature. However, due to their geometry, the influence of FBG on wave propagation is assumed to be negligible.

In contrast, the integration of discrete sensors requires more profound research and can be divided into attempts for single sensors and for sensor networks. Single sensors can be integrated quite well without affecting their performance and without weakening the structure substantially. This is investigated for three-point bending and fatigue tests and is confirmed in the presence of damage to the structure [30,31]. Another approach is the integration of full built-in sensor networks using flexible PCB layers with an applied network. They can be used as a whole layer during laminate layup [32,33].

To evaluate a monitoring system with integrated sensors, the influence on the mechanical properties has to be taken into account. An abundance of research addresses the integration of lead zirconate titanate (PZT) sensors in carbon fiber-reinforced polymer (CFRP). The integration of single sensors shows no influence on the fatigue strength [34], but the tensile and compressive modulus are reduced by approx. 21% and 13%, respectively [35]. A polyimide insulation layer reduces the compressive, flexural and interlaminar shear strength in CFRP specimens whereas the usage of small glass fiber-reinforced polymer (GFRP) samples as insulation does not [36]. The integration of a whole PCB layer with a PZT transducer network in CFRP only minimally influences the material properties of the laminate such as shear strength [33]. An investigation with graphite/epoxy laminates shows no effect on the tensile strength and Young’s modulus when PZT actuators are embedded [37].

Although the integration of sensors shows no significant degradation of the structure’s mechanical properties, a difference in performance between surface-bonded and embedded transducers might occur. A comparison of emitter–receiver combinations in GFRP shows that there are configurations where embedding both the emitter and receiver provides the best signals, even compared to two surface-bonded transducers [38]. This behavior seems to be strongly dependent on the configuration of material, layup and frequency used, as it has been demonstrated that sensors integrated into CFRP show higher signal amplitudes for surface-mounted emitting transducers than for embedded ones [39]. This is supported by a further investigation on an integrated sensor network into CFRP. It demonstrates that the integrated sensor network shows little environmental noise as well as stable and repeatable data acquisition characteristics in comparison with surface-applied PZT sensors [40].

Experimental setups used to determine wave characteristics need well-defined boundary conditions and a wave field with as little disturbances as possible. Normally, the largest possible specimens are chosen to avoid reflections from the edges and their superposition with the signal of interest. However, large specimens increase manufacturing effort, costs and material consumption. Narrow specimens avoid these drawbacks while showing a more complex wave field. The suitability of strip-like specimens to investigate GUW propagation is subject to current research. An investigation looked at whether a narrow isotropic aluminum specimen could be simplified as a 1D structure and used to generate a straight wave front. The propagation of flexural and axial waves was numerically simulated in a 14 mm wide strip for the investigation of crack interaction. It showed straight wave fronts along the wave propagation direction in addition to an increasing dispersion and diminishing coherence when the signals superimpose with reflections from the strip’s end [41]. Increasing the specimen’s width to, e.g., 40 mm leads to amplitude variations along and across the specimen width indicating the presence of standing waves due to the superposition of reflected waves from the specimen’s edges [42]. One strategy to cope with this challenge is to extract the relevant wave packages from the reflections coming off the specimen’s edges (time gating) [43].

In summary, on the one hand, the literature review presents investigations on GUW interaction with damage. On the other hand, the performance of integrated discrete sensors for SHM applications using GUW is investigated. However, the influence of the sensor and corresponding electrical contacting integration on the wave field has not yet been sufficiently addressed. In addition, the geometry of a possible sensor is of interest.

The aim of this paper is to investigate the influence of an embedded sensor that is presented by the authors [9] and mounted on a polyimide-based PCB, on the propagating GUW field using narrow GFRP strip specimens to reduce the manufacturing effort.

This work shows that the integrated inlays cause reflections that cannot be experimentally detected and are therefore assumed not to influence the wave propagation significantly. However, reflections from the specimens’ edges do occur and superimpose with the propagating waves. In this study, a method is presented to distinguish between edge reflections and the interactions at integrated inlays using the periodic behavior of the occurring wave packages. This method includes a spatial continuous wavelet transformation to identify the different kinds of reflections in the time–space domain of the transformation and to assign the observed wave packages to specific wave modes.

Hence, this article is structured as follows. The materials and methods section starts with the design of the GFRP specimen with an integrated sensor inlay and an estimation of the related dispersion relation. The numerical model to deduce possible wave interactions between GUW and the inlay is presented. This is followed by an introduction of the experimental setup of the air-coupled ultrasonic technique and a 3D laser Doppler vibrometer (LDV) measurement. The methods section closes with a description of the continuous wavelet transformation (CWT) used to evaluate the experimental data. Subsequently, the numerical and experimental results are presented. Regarding the extracted B-scans of the measurement, possible reflections in the wave field are discussed. Subsequently, the CWT is used for time signal reconstruction to identify reflected wave packages. The work closes with a discussion of the results including a critical analysis of the experimental procedures and a comparison between the numerical and experimental results.

## 2. Materials and Methods

### 2.1. Design of the Specimens

Considering the possible integration of sensors into plate-like structures for SHM using GUW, a simplified approach is followed. As already mentioned, the sensor investigated is a MEMS vibrometer composed mainly out of borosilicate glass [9]. As no sensing effect is required to study the effect of the sensor on the wave field, the sensor can be simplified by a bulk glass body constructed from the same material as the MEMS vibrometers. In real-world applications, an electrical contacting of the sensor is needed. PCBs based on polyimide substrates with copper conducting traces represent the state of the art for contacting electrical components. Therefore, the representative samples under investigation consist of a glass body (cf. Figure 1), representing the discrete sensor [9], which is adhesively bonded to a polyimide substrate, representing the PCB, using a high-temperature-resistant cyanoacrylate, a typical bonding material.

Following this procedure, six different strip specimens are manufactured. Figure 2 illustrates an example of a strip specimen with polyimide inlay and applied square sensor body in plane and cross-sectional view. The strip specimens’ height are approx. 2 mm comprising 16 prepreg layers with a nominal thickness of 0.12 mm each. Out of the six specimens, five are integrated with a polyimide inlay. The polyimide inlay is 200 mm long, 2 mm wide and 25 μm in thickness. Among these five specimens, four additionally have differently shaped sensor glass bodies adhesively bonded on the end of the polyimide inlay using cyanoacrylate. The cross section of the sensor bodies in the wave propagation direction is 2 mm in width and 200 μm in height. The different sensor shapes are shown in Figure 1. The aim is to investigate whether the sensor shape also influences the wave propagation behavior. For GUW excitation, a PZT ceramic actuator is applied 100 mm away from the beginning of the inlay.

An overview of the specimens’ inlay specifications is given in Table 1. A strip specimen example is shown in Figure 3.

For the design of the strip specimens, we employed the unidirectional GFRP prepreg DLS1611 from Hexcel Corp. The fiber orientation is chosen to be perpendicular to the strip specimen’s length since the lower stiffness in this direction leads to smaller velocities. The aim is to selectively reduce the propagation velocities in the direction of the strip specimen’s length by design. The dispersion relation for a laminate setup offers information about the expected modes’ phase velocities as shown in Figure 4a. This represents the specific propagation speed of a Lamb wave mode at a specific frequency. The dispersion relation also provides information about the wavelengths of the wave modes per frequency as shown in Figure 4b. This information is used later for wave mode identification. Hence, regarding the analytically solved dispersion relations shown in Figure 4a,b, the fundamental wave modes are expected to have a minimum wavelength of 3 mm and a maximum phase velocity of 2200 m/s. The material parameters assumed for the solution of the dispersion relation are standard values for unidirectional GFRP provided in Table 2.

### 2.2. 2D Numerical Model to Simulate GUW Interactions with Integrated Polyimide Inlays

To analyze the impact of sensor inlays on GUW propagation without any experimental side effects, numerical simulations are first conducted as a preliminary investigation. The numerical simulations are not meant to represent the subsequent experiments in their entirety, but to instead provide a qualitative impression on what to expect during the LDV and air-coupled measurements.

For the finite element simulation, a two-dimensional model, cf. Figure 5, under plane strain assumption is used that represents the cross section of the specimen introduced in Section 2.1. To investigate the influence of the polyimide inlay on wave propagation, two different waveguides are used. The first model represents a GFRP specimen without an inlay to observe the wave propagation in an undisturbed cross section. Following the stacking sequence provided in Section 2.1, the size of the numerical model is 500 mm × 1.92 mm with the fibers orientated perpendicular to the wave propagation direction. The second model includes the polyimide inlay, indicated in green in Figure 5. The work presented here focuses on the linear wave propagation. Therefore, inhomogeneities are only detectable when their size falls at least within the magnitude of the wavelength [10]. Consequently, the sensor body itself is not incorporated into the modeling procedure because its overall size of 2 mm × 0.2 mm is smaller than the expected wavelength of the $A_{0}$-wave mode. The material properties of the components are provided in Table 2.

In the numerical simulation, the wave field is excited by a five-cycle Hanning windowed sine burst with a center frequency of 75 kHz, 120 kHz, and 200 kHz, respectively. The same signal is later used for the subsequent LDV-measurement. The excitation is realized by an out-of-plane displacement at the upper-left corner of the numerical model. Furthermore, a symmetry boundary condition is applied to the left edge, cf. Figure 5. The distance between the excitation and the polyimide inlay is selected in such a way that the two fundamental modes $A_{0}$ and $S_{0}$ can be analyzed separately to gain a better understanding of the wave interaction with the inlay. All remaining boundary conditions are depicted in Figure 5.

To ensure correct simulations of the wave propagation in the composite, a sufficient spatial discretization $\Delta x_{\max}$ and time stepping $\Delta t_{\max}$ must be selected to represent the minimally occurring wavelength $\lambda_{\min}$. Here, the following conditions apply [14,44]
(1)$$\Delta t_{\max}=\frac{1}{20f_{\max}},$$
(2)$$\Delta x_{\max}=\frac{\lambda_{\min}}{20}.$$

However, the discretization of the model depends not only on the wavelength of the wave modes, but also on the layered structure of the waveguide. Due to the thin polyimide layer relative to the overall specimen thickness, the selected element edge length is much smaller than the required value following Equation (Equation 2). Furthermore, a symmetric discretization is targeted. Therefore, the inlay is discretized with 2 elements over the thickness, whereas a total of 40 elements are used for the GFRP layers. This leads to an almost equal vertical edge length of the elements over the thickness of the specimen. To further ensure an almost square shape of the elements, 10,000 second-order Lagrange elements are used alongside the specimen. Hence, in total 400,000 9-node-elements are used to form a structured mesh of the waveguide.

The numerical simulations are conducted in COMSOL Multiphysics. With the presented discretization, the computation time of the wave propagation in GFRP with a polyimide inlay takes approximately 6 h.

### 2.3. Experimental Setups for Determination of GUW Interactions at Integrated Inlays

To analyze the detectability of the wave interaction with the polyimide inlay in real specimens, two independent experimental procedures are performed. As in the numerical simulations, B-scans are extracted as a representation of the wave field along the path from the actuator to the inlay. This method is performed for both experiments. In the first run, the wave propagation is measured using an air-coupled ultrasonic technique. All six specimens are examined with regard to a possible reflection phenomenon at the polyimide inlay. In a second run, a 3D LDV measurement is used at different excitation frequencies for one specimen. The two different measurement procedures allow the experimental separation and evaluation of the different displacement field components. Details of the two procedures are described in the following.

#### 2.3.1. Detection of GUW Interactions Using Air-Coupled Ultrasonic Technique

The air-coupled ultrasonic technique is an established method in non-destructive testing and the setup in this study is exemplary, shown in Figure 6. It uses an applied actuator on the structure’s surface instead of a test probe [8]. The frequencies of the structure’s sound radiation can be detected by microphones which are sensitive to a narrowband frequency range and need to be chosen with reference to the structure’s excitation frequency. For the experiments presented, a USPC 4000 AirTech (Hillger NDT GmbH, Braunschweig, Germany) is used. As in the numerical simulations, the excitation frequencies are 75 kHz, 120 kHz and 200 kHz. Following this, the microphones employed for the detection are AirTech 75, AirTech 120 and AirTech 200 (Hillger NDT GmbH, Braunschweig, Germany).

A round PZT ceramic (material: PIC255, diameter: 16 mm, thickness: 0.2 mm, type: PRYY-1126, PI Ceramic GmbH, Lederhose, Germany) is glued to the specimen using Loctite EA 9466 (Henkel AG & Co. KGaA, Düsseldorf, Germany) and used as an actuator. The excitation signal is a three-cycle Hanning-windowed rectangular burst signal with the named frequencies for all six strip specimens. Although it shows a wider bandwidth of excitation frequencies, it contains the same center frequency as a sinusoidal burst and hence, gives comparable signals. Increasing the excitation frequencies leads to a decrease in the amplitude of the measured signals due to material damping effects. Therefore, the amplification of the excitation signals must be adjusted. For the three measurements, excitation voltages from 35.5 V to 50.2 V are used.

The measuring path is presented in Figure 3, with a length of 235 mm and a spatial resolution of 1 mm, resulting in 5 points per minimally occurring wavelength, cf. Figure 4b. The beginning of the polyimide inlay is located at a distance of 100 mm, where possible reflections are to be expected.

The recorded microphone signals are filtered by a bandpass of 12th order with a bandwidth from 0.5–1.5 times the burst center frequency using receiver amplification factors from 31.5 dB to 43.5 dB using a preamplifier AirTech 4026 (Hillger NDT GmbH, Braunschweig, Germany). All measurements are averaged eight times.

Variations in probe angle allow for the detection of different components of the excited wave modes. Therefore, two probe angles are chosen: probe perpendicular to the strip specimens and a probe angle of approx. 10° to the vertical. Using the probe perpendicularly allows the detection of the pure out-of-plane component of the displacement field, while using the probe under a certain angle detects a superposition of out-of-plane and in-plane components of the propagating waves.

#### 2.3.2. Detection of GUW Interactions Using 3D Laser Doppler Vibrometry

The aim of the 3D LDV measurement is the extraction of the in-plane and out-of-plane components of the wave propagation to allow separate evaluation [4] and better comparability to the numerical results. The setup featuring a PSV-400 vibrometer (Polytec GmbH, Waldbronn, Germany) is depicted in Figure 7.

The excitation signal is a five-cycle Hanning-windowed sine burst signal that is amplified by a high-voltage amplifier WMA-300 (Falco Systems, Katwijk aan Zee, The Netherlands). In reference to the simulation described in Section 2.2 and the air-coupled-measurements described in Section 2.3.1, the excitation frequencies for the experiment are 75 kHz, 120 kHz, and 200 kHz.

Prior to the investigations, a rectangular actuator (material: PIC255, width: 50 mm, length: 10 mm, thickness: 0.2 mm, PI Ceramic GmbH, Lederhose, Germany) replaces the round PZT ceramic on the strip specimen with the polyimide inlay without additional sensor body according to Figure 3. The rectangular ceramic is selected to avoid early reflections in the wave field due to the radially symmetric wave fields propagating from the round PZT ceramics. The intention is to create a straight wave front propagating in the strip specimen before superimposing with reflections from the specimens’ edges.

The measurement path is depicted in Figure 3. The beginning of the polyimide inlay is located at a distance of 90 mm. This means that possible reflections would occur at a path length of 90 mm in the later extracted B-scans. A spatial resolution of 0.44 mm is selected, which corresponds to approximately ten points per minimally expected wavelength, cf. Figure 4b.

Retroreflective tape is applied to the specimen’s surface along the measurement path to increase the signal quality of the LDV measurement. Three subsequent measurements are performed at different angles and a coordinate transformation is implemented to decompose the measured surface velocities into one out-of-plane and two in-plane components [4,45].

### 2.4. Identification of Occurring Reflections by Means of Continuous Wavelet Transformation

In the results section, a clear identification of wave packages according to their occurrence in space and time by means of examining their wavelength is necessary. Within this work, a continuous wavelet transformation is used for this purpose.

A wavelet transformation offers the possibility to extract information about the occurring frequencies in a time signal over a localization in the time-domain. Hence, it is a method of direct time-frequency analysis, which works using a time-limited window with an average amplitude of zero and a variable size [2]. A single wavelet function is the basis for wavelets constructed by translation and dilation. The location of the wavelet in the time domain is defined by a scaling and a translation parameter [47]. The transformation’s resolution at different times and frequencies is governed by the Heisenberg uncertainty principle [47]. There are a variety of wavelets that are selected depending on the application and the time signal’s characteristics.

The CWT of a time signal leads to a representation of the energy distribution per frequency in the time-domain [2]. Since the discussed signals in this work are not continuous but change the instantaneous frequency within a certain range of time, a fixed window-size, e.g., in the short-time Fourier transformation (STFT), might lead to a loss of information. In comparison to STFT, CWT offers a variable window size and thus higher precision simultaneously in both the time- and frequency-domain [2]. A CWT can also be undertaken in the spatial domain, which leads to occurring wavenumbers over the spatial resolution instead of occurring frequencies over the time-scale, a feature that will be extensively used in the evaluation of the experimental data in Section 3.4.

In this study, the experimental time signals are filtered using CWT in the time-frequency domain. They are reconstructed using the inverse CWT with a narrower frequency bandwidth set to ±5 kHz around the burst center frequencies to remove discontinuities causing dissipation in the spectra. This helps to identify wave packages better. Afterwards, the spatial CWT is performed to identify where the different wavelengths corresponding to the two fundamental modes occur spatially. The identification of wave packages can be deduced by the appearance of a specific mode’s wavelength at a certain position in space and time, cf. Section 3.4. These steps are conducted for the measurements of all three frequencies and the temporal and spatial CWT are performed using the analytic Morse wavelet. The symmetry parameter is set to 3 and the time-bandwidth product is 60. A total of 10 voices per octave are used [48].

## 3. Results

### 3.1. GUW Interactions at Integrated Polyimide Inlays Estimated by the 2D Numerical Model

The results of the numerical simulations are provided in Figure 8 and Figure 9. To generate the depicted B-scans, the time signals are vertically aligned for each observation point along the specimen. The result is a surface plot, where the amplitude of the wave propagation is plotted over the propagation direction (horizontal axis) and time (vertical axis). Figure 8 presents B-scans of the in-plane and out-of-plane displacement component for the wave propagation in a pure GFRP waveguide (Figure 8a,c) and a waveguide with a polyimide inlay (Figure 8b,d) at 120 kHz. In both cases, two wave modes are excited, the fundamental $S_{0}$- and $A_{0}$-mode. The divergence of the $A_{0}$-mode with the propagation distance is caused by its strong dispersive nature. This can be derived from the dispersion diagram, cf. Figure 4a. Comparing the in-plane and out-of-plane displacement component for the GFRP waveguide reveals that the in-plane displacement component of the $S_{0}$-mode is dominant over the out-of-plane component, whereas for the $A_{0}$-mode the out-of-plane component is dominant. This fits the characteristics of GUW in thin-walled structures [49]. Inserting an inlay into the structure at x= 30 mm, cf. Figure 5, causes reflections of the two wave modes at the impedance change. First, the $S_{0}$-mode passes the change of the cross section, which leads to a partial reflection of the wave mode, indicated by an additional wave package which propagates back to the excitation point with the same absolute slope. However, this is only detectable in the in-plane displacement field. For the out-of-plane motion, the interaction is barely observable. In the upper left corner, a slight disturbance is recognizable in the $A_{0}$-mode, which meets the characteristics of the partially reflected $S_{0}$-mode in Figure 8b. After the $S_{0}$-mode passes the cross section change, the $A_{0}$-mode interacts with the inlay. The partially reflected $A_{0}$-mode is clearly detectable in both displacement components.

To gain a deeper insight into the influence of the inlay on wave propagation, Figure 9 shows the results of numerical simulations at the remaining frequencies of 75 kHz and 200 kHz. Again, both displacement field components are provided for each simulation. The same conclusion can be derived from Figure 9a,b (75 kHz) as well as Figure 9c,d (200 kHz). Beside the signal at 75 kHz, the interaction of both fundamental wave modes is clearly detectable in the in-plane displacement component, whereas the out-of-plane component reveals only partial reflection of the $A_{0}$-mode. For 75 kHz, due to the high wavelength, the signal is temporally not long enough to also capture the reflected $A_{0}$-mode.

In conclusion, an interaction between the excited GUW and the polyimide inlay is found for a certain frequency range. Both the fundamental $A_{0}$- and $S_{0}$-mode interact with the inlay. However, it is important to note that the resulting amplitude is more than one order of magnitude smaller than the amplitudes of the excited wave modes. For this reason, the B-scans are plotted with a logarithmic amplitude scale. For experimental validation, the main focus is set on the in-plane displacement since the interaction of the $S_{0}$-mode is properly detectable.

### 3.2. Detectability of GUW Interactions Using Air-Coupled Ultrasonic Technique

Measurements by means of the air-coupled ultrasonic technique as presented in Section 2.3.1 are performed to experimentally validate the theoretical results presented in Section 3.1.

Exemplary for all measurements and in reference to the numerical simulations described in Section 3.1, the following B-scans are presented in this paper: strip specimen with polyimide inlay at 75 kHz with perpendicular probe in Figure 10a, strip specimen with polyimide inlay at 120 kHz with perpendicular probe in Figure 10b, and strip specimen with polyimide inlay at 120 kHz with a probe angle of approx. 10° in Figure 10c. As visible in Figure 10c, two wave fronts occur which means that under an angle of the probe both the slower $A_{0}$- and the faster $S_{0}$-mode can be detected. In the B-scans of the measurements with the perpendicular probe, only the $S_{0}$-mode can be detected. In total, 36 scans are recorded for three excitation frequencies, two probe angles, and six strip specimens.

The measurement path described in Figure 3 shows that reflections are expected to occur in the B-scans at a location of 100 mm. It is evident that the air-coupled technique measurements in Figure 10 do not show any visible reflections at the polyimide inlay. The measurements in the other specimens show the same result whether a glass sensor body is applied or not. This finding is independent from whether the scans of the waves’ displacement field contain pure out-of-plane wave motion or superimposed in-plane and out-of-plane motion. This leads to the conclusion that a separate investigation of the in-plane and out-of-plane component of the wave field is necessary to allow a more accurate comparison with the simulations. Additionally, the glass sensor body is neglected for the following investigations as referring to the model assumptions in Section 2.2, the dimensions of the sensor are too small to influence wave propagation with wavelengths larger than the sensor length.

### 3.3. Detectability of GUW Interactions Using 3D Laser Doppler Vibrometry

To separately evaluate the in-plane and out-of-plane component of the wave propagation, the results of the 3D LDV measurement as described in Section 2.3.2 will be discussed in the following.

Figure 11a–c hold the B-scans of the in-plane velocity component of the wave propagation. A logarithmic amplitude scale is selected to visualize even small proportions of the wave propagation. The fundamental $A_{0}$- and $S_{0}$-modes can be identified in all three B-scans.

The wave propagation speed of the $S_{0}$-mode can be derived from the slope in the B-scan as no significant dispersion occurs in this frequency range, cf. Figure 4a. As shown in Table 3, wave speeds could be identified that suit well the theoretically determined phase velocities in Figure 4a and indicate a slightly decreasing velocity with an increasing frequency regarding the $S_{0}$-mode. Deviations between experimental and simulative results are mostly due to the assumption of erroneous material properties in the dispersion diagram generation as no material data sheet from the manufacturer is available.

It can be seen in all figures that no reflections at the polyimide inlay can be identified. Instead, wave packages occur that cannot be directly linked to the propagating $A_{0}$- and $S_{0}$-modes. In addition, the location of their occurrence cannot be explained by reflections at the specimens’ end. The amplitudes of these reflections occur in the order of magnitude of the $S_{0}$-mode and the slope of the wave packages suggests that the reflection is another $S_{0}$-mode. At 200 kHz, four wave packages can be identified showing the velocity of the $S_{0}$-mode. They are marked with white circles in Figure 12a. An overview of their precise position in the spatial and time domains is given in Table 4. The first wave package occurs due to near field effects near the actuator, since it is slightly shifted compared to the other three wave packages. The three wave packages afterwards show a clear periodic behavior without any shift.

The wave packages’ variation in amplitude along the B-scans can be explained by the periodic superposition of back and forth running waves between the narrow edges of the specimen causing standing waves to superpose the GUW. This will be verified in the following section using a continuous wavelet transformation as described in Section 2.4 to distinguish these edge reflections from possible reflections at the polyimide inlay.

### 3.4. Identification of GUW Interactions by Means of Spatial Continuous Wavelet Transformation

In this section, a spatial CWT is used to identify the occurring wave packages described in Section 3.3 and to distinguish them from possible reflections at the embedded inlay. The in-plane component of the displacement field in wave propagation direction perpendicular to the fibers is selected from the 3D LDV measurement as it shows best reflection visibility in the numerical results in Section 3.1. Since the $S_{0}$-mode shows a higher in-plane component than the $A_{0}$-mode, the former is used for further evaluation. Referring to Section 2.3.2, the inlay-related reflections are expected to be at a position of 90 mm in the B-scan. However, the B-scans in Section 3.3 do not show reflections at this position. For evaluation of the experimental data, temporal and spatial CWT are performed as described in Section 2.4.

We investigated whether the wavelength of the $S_{0}$-mode appears in the identified periodically occurring wave packages as this would confirm reflections of the fast $S_{0}$-mode from the strip specimen’s width. For this purpose, for each excitation frequency, the first reflected wave package is localized in space and time in the B-scan. Three points in time are selected from the measurement located at the center of the wave package: a time before the reflected wave package occurs, one in the wave package’s center and one after the wave package has decreased. This is shown for 75 kHz in Figure 12b. An overview of the times and locations for all three excitation frequencies is given in Table 5.

Due to the multimodal and dispersive behavior of the GUW, it is expected that the transformed signals in the wavelength–space-domain will not appear in clear delineation.

In the following, a complete analysis for the measurement at 75 kHz will be presented. Additionally, the wavelengths at one point in time will be analyzed in the center of the first identified reflected wave package for 120 kHz and 200 kHz as listed in Table 5.

#### 3.4.1. Reflection Identification at 75 kHz

The results of the spatial CWT for 75 kHz and the points in time defined in Table 5 are shown in Figure 13a–c. According to Figure 12b, a location of 40 mm is under investigation, cf. dashed white line in Figure 13a. The spectra obtained using the CWT for all three points in time are investigated for occurring wavelengths corresponding to the two fundamental wave modes.

At 40 mm and 50 μs, the $S_{0}$- and $A_{0}$-modes start to diverge. At this location and this point in time, no reflections occur yet, cf. Figure 12b, and both wave modes are shown in the spatial CWT as derived in Figure 13a. Two horizontal peak branches are visible at wavelengths of approx. 10 mm and at approx. 30 mm. The occurring wavelengths fit well to the numerically derived values indicated by horizontal dashed white lines.

Figure 13b shows the CWT of the time signal in the temporal center of the reflection. According to the group velocities, the $S_{0}$-mode should have already passed. However, two low intensity branches can be identified for wavelengths corresponding to both the $S_{0}$- and $A_{0}$-modes. As the $S_{0}$-mode appears as well, this indicates the presence of an $S_{0}$-mode reflection at the strips edges.

Figure 13c depicts the CWT for a point in time after the reflected wave package. A high-intensity branch at a wavelength of approx. 12mm, corresponding to the $A_{0}$-mode, occurs and remains visible at 40mm. However, the $S_{0}$-mode branch at approx. 30mm does not occur which confirms the non-appearance of the $S_{0}$-mode after the periodically appearing reflection wave package.

**Figure 13 materials-15-06752-f013:**
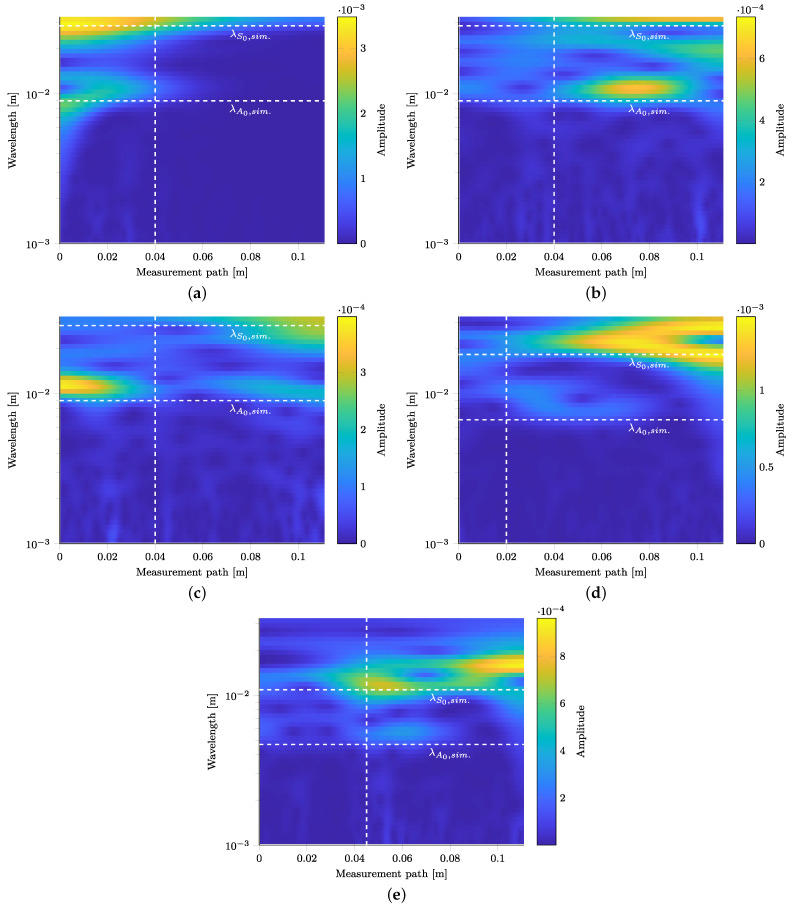
Spatial CWT of the 3D LDV measurements at time steps as stated in Table 5. Horizontal dashed lines indicate the wavelengths λ from the dispersion diagrams. Vertical dashed lines indicate locations of interest of identified wave packages in the B-scans. (**a**) 75 kHz and 50 μs (cf. Figure 12b). (**b**) 75 kHz and 115 μs (cf. Figure 12b). (**c**) 75 kHz and 145 μs (cf. Figure 12b). (**d**) 120 kHz and 75 μs (cf. Figure 11b). (**e**) 200 kHz and 70 μs (cf. Figure 11c).

#### 3.4.2. Reflection Identification at 120 kHz

At an excitation of 120 kHz, the first reflected wave package in the B-scan of the in-plane component is identified at 20 mm and 65 μs to 85 μs. In this time range, an appearance of the $S_{0}$-mode is not expected because the propagating mode has already passed the location and the $A_{0}$-mode starts to appear. As visible in Figure 13d, the CWT at 75 μs shows both wave modes in two branches over a wide spatial range. The continuous appearance of the $S_{0}$-branch is an indication for reflections from the strip specimen’s width. The amplitude maximum at the end of the strip specimen can be explained by the excited wave package passing by before becoming reflected at the strip specimen’s end.

#### 3.4.3. Reflection Identification at 200 kHz

As shown in Figure 12a, the center of the first periodically repeating wave package that is not due to near field effects is located at approx. 45 mm. The center of the wave package in the time domain is at 70 μs and it is not to be expected that the wavelength of the $S_{0}$-mode occurs in the spatial CWT. However, a clear intensity peak of the $S_{0}$-mode occurs in Figure 13e at the investigated spatial position. Therefore, the occurrence of significant reflections of the $S_{0}$-mode is evident.

For all frequencies, a spatial CWT can be used to show that the periodically occurring wave packages have the wavelength of the $S_{0}$-mode. A slight increase in wavelength can be explained as the straight wave fronts no longer occur due to reflections from the specimen’s edges. Different angles of incidence through the measuring path occur and distort the waves in the analysis. Enclosed areas of vanishingly small amplitude in the spatial CWT can be explained by destructive interference phenomena by superposing reflections. Taking into account the wave packages’ location in space and time, this finding proves the superimposing reflections from the specimens’ edges and that these can be distinguished from possible reflections at the polyimide inlay.

## 4. Discussion

This section discusses the methods and results presented above to assess whether reflections at the integrated polyimide inlays occur and affect the wave field.

The numerical simulations indicate that reflections of the $S_{0}$-mode theoretically occur at the inlay, but are at least an order of magnitude smaller than the excited $S_{0}$-mode. This is an initial indication that the polyimide substrates lead to only minor reflections that might not be dominant in experimental investigations.

However, independently of the sensor body’s shape, no reflections are detected by the air-coupled ultrasonic technique. In contrast to the polyimide inlay, the sensor bodies representing the MEMS vibrometers under investigation [9] are smaller than the occurring wavelengths in the experiment. Thus, it is reasonable that any possible sensor body shape applied to the polyimide inlay presented can be neglected for later GUW field experiments in the specimens presented when assuming the concept of linear wave propagation.

Although the numerical simulation predicts the highest reflection amplitude for the in-plane component, this cannot be validated experimentally. By extracting the corresponding motion with a 3D LDV measurement, no reflections are detected in specimens with embedded polyimide inlays. It is concluded that material damping effects of the epoxy resin occur to such an extent that reflections in the experiments are attenuated below the detection threshold.

Despite using a rectangular PZT actuator, periodically appearing wave packages are detected. However, by performing a spatial and temporal CWT, they can be identified as $S_{0}$-mode reflections from the specimen’s edges. This enables a clear distinction from possible inlay reflections.

Therefore, regarding the narrow GFRP specimens at hand, the GUW experiments and methods used can be successfully performed and reveal no detectable GUW reflections at the integrated polyimide inlays.

This leads to the conclusion that polyimide inlays are suitable as PCB substrates for integrating sensors such as MEMS vibrometers [9] into FRP for SHM using GUW without significantly influencing the wave propagation.

## 5. Conclusions

The aim of this work is to investigate the influence of integrated sensor substrates on the GUW propagation in FRP specimens. Here, a thin and narrow polyimide inlay serves as the substrate for a PCB with applied representation of a MEMS vibrometer [9].

Initial numerical simulations in an undamped model show reflections at an amplitude of at least an order of magnitude smaller than the incident wave. In a 3D LDV experiment, reflections from the specimens’ edges can be clearly identified with a spatial-temporal CWT due to their localization in space and time. Since they show a clear periodical behavior, they can be distinguished from possibly detectable inlay reflections which leads to the conclusion that polyimide substrates can be used for PCBs to integrate GUW sensors inside FRP without influencing the wave propagation in a detectable manner. As a main advantage of this study, reflections from the specimens’ narrow edges can be clearly identified using easy-to-implement signal transformation methods. However, conclusions on other laminate structures and strip geometries require further investigation.

Future studies will investigate the extent to which other variables influence the results obtained in this work. Possible investigations can be performed regarding different fiber materials, i.e., 3D woven fabrics, fiber orientations, laminate setups, ambient temperatures, polyimide thicknesses and different sensor materials. Further investigations within the framework of a parameter study are being sought. The sensor is to be accounted for in the numerical model and the simulation results are to be compared.

## Figures and Tables

**Figure 1 materials-15-06752-f001:**
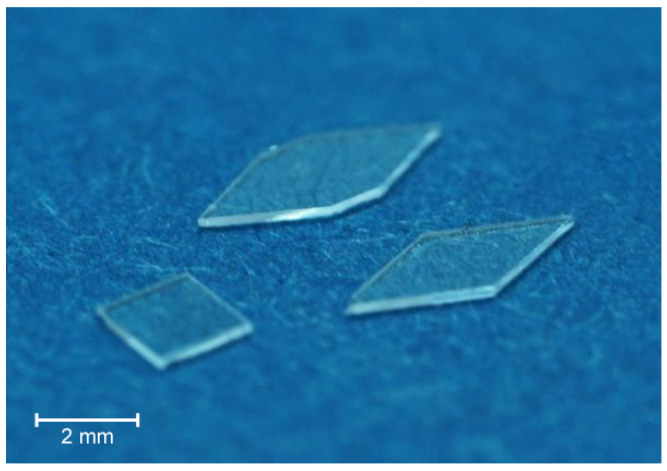
Glass sensor bodies as dummies for simulating the integration of a MEMS vibrometer [9] into a laminate. The glass inlays are constructed from BOROFLOAT33 borosilicate glass wafers with a thickness of 200 μm and shaped using a wafer-dicing saw.

**Figure 2 materials-15-06752-f002:**
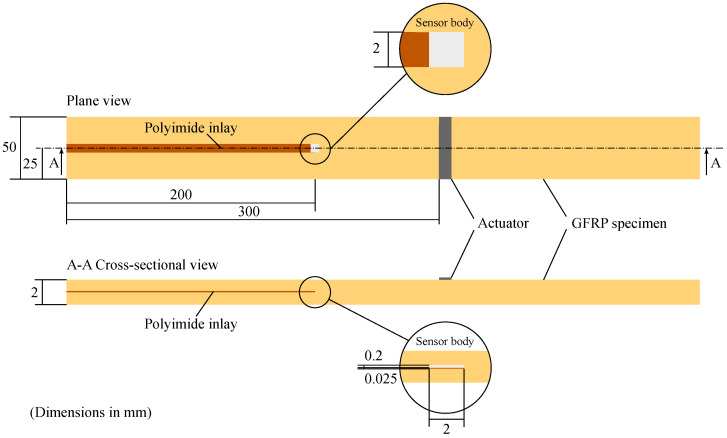
Sketch of the GFRP specimen design showing the dimensions, the polyimide inlay and an applied square dummy sensor body (cf. Figure 1) composed of the same borosilicate glass as the representative MEMS vibrometer [9] under investigation.

**Figure 3 materials-15-06752-f003:**
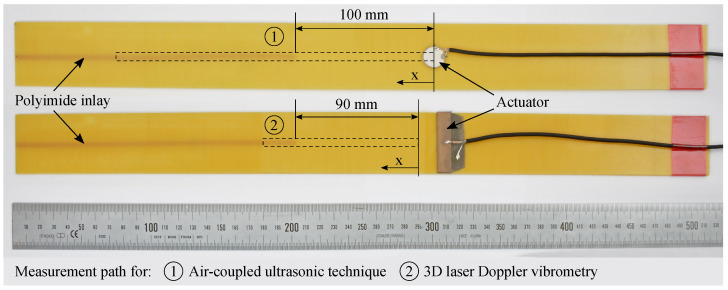
GFRP strip specimens with scale, visible polyimide inlay (orange, left) as well as rectangular (**top**) and circular (**bottom**) PZT actuators. For the exact dimensions and setup cf. Figure 2. In addition, the measurement paths and coordinates for the measurements described in Section 2.3 are depicted.

**Figure 4 materials-15-06752-f004:**
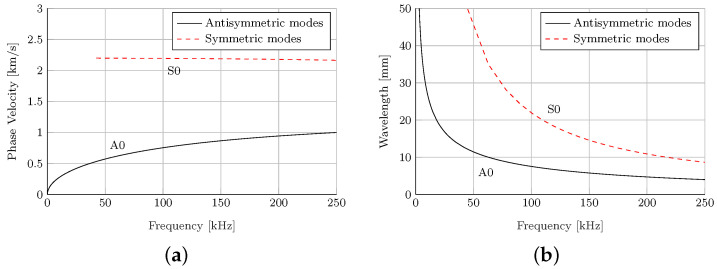
Dispersion diagrams of the GFRP specimen under investigation. Material properties used can be found in Table 2. (**a**) Phase velocities of the fundamental modes. (**b**) Wavelengths of the fundamental modes.

**Figure 5 materials-15-06752-f005:**
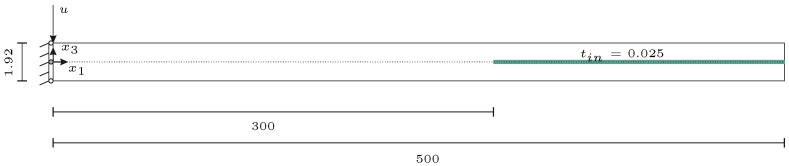
Sketch of the two-dimensional numerical model used to simulate the interaction between the GUW propagation and a polyimide inlay (green) integrated into a GFRP laminate. All measurements are in mm.

**Figure 6 materials-15-06752-f006:**
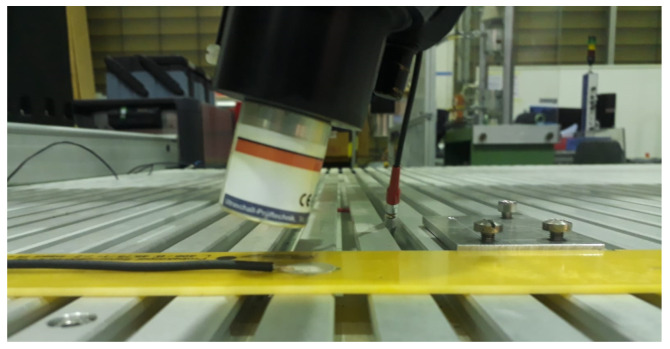
GFRP strip specimen with attached actuator and microphone above for detecting GUW propagation using air-coupled ultrasonic technique. The microphone depicts a variation of the probe angle. The inclination additionally enables the detection of in-plane proportions compared to a perpendicular setup, which is only sensitive to out-of-plane components.

**Figure 7 materials-15-06752-f007:**
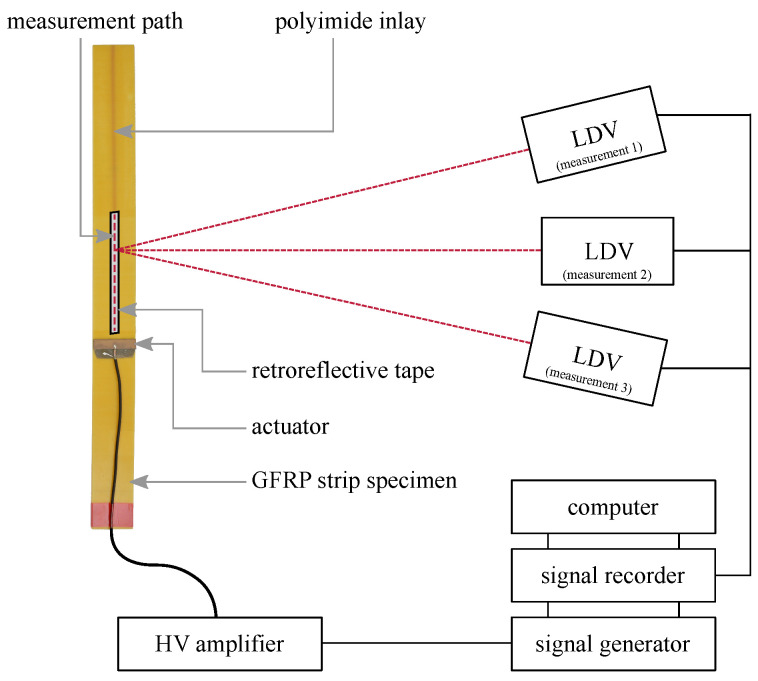
Experimental setup of 3D laser Doppler vibrometry measurement to separately determine in-plane and out-of-plane components of wave propagation after performing a coordinate transformation according to [45]. Figure adapted from [46].

**Figure 8 materials-15-06752-f008:**
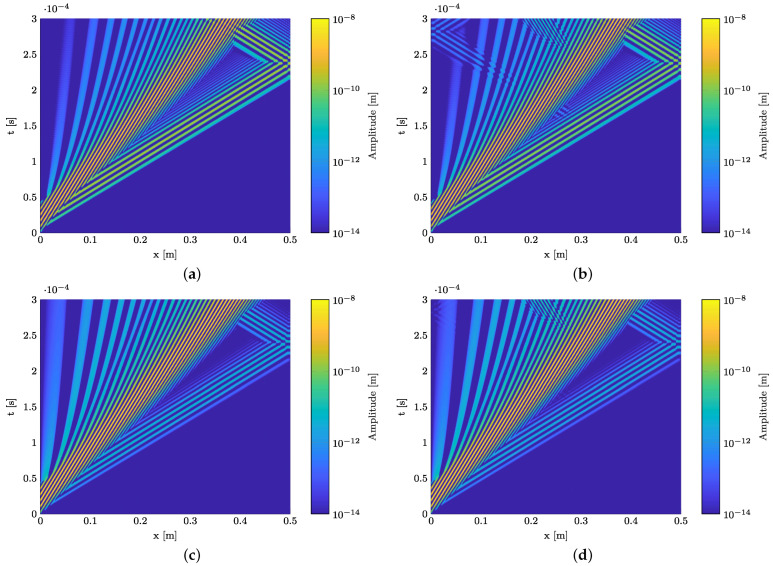
Comparison of the numerically determined wave propagation in a pure GFRP laminate and a waveguide including a polyimide inlay in the midplane at a burst center frequency of 120 kHz using a B-scan representation. The polyimide inlay starts at 0.3 m (cf. Figure 5). (**a**) Wave field without polyimide inlay, in-plane displacement field. (**b**) Wave field with polyimide inlay, in-plane displacement field. (**c**) Wave field without polyimide inlay, out-of-plane displacement field. (**d**) Wave field with polyimide inlay, out-of-plane displacement field.

**Figure 9 materials-15-06752-f009:**
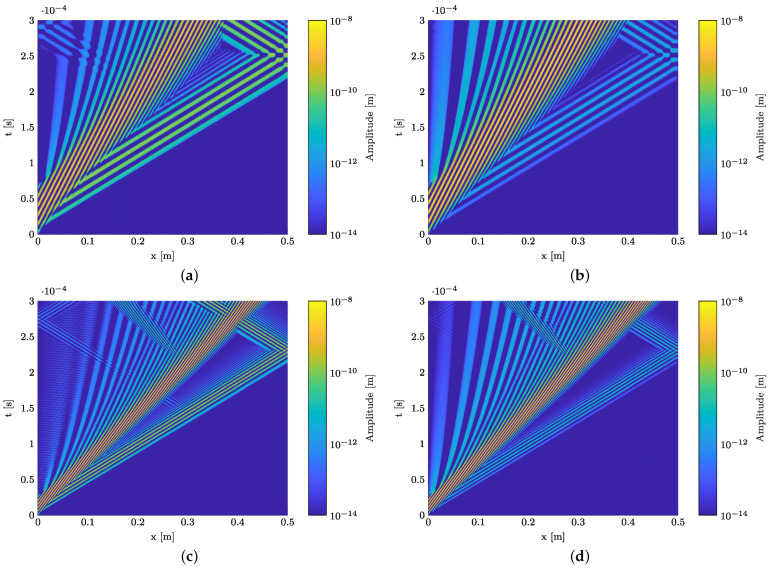
Comparison of the numerically determined wave interaction of GUW with the polyimide inlay in the midplane of the GFRP specimen at different burst center frequencies. The design of the numerical model is depicted in Figure 5. (**a**) In-plane displacement component at 75 kHz. (**b**) Out-of-plane displacement component at 75 kHz. (**c**) In-plane displacement component at 200 kHz. (**d**) Out-of-plane displacement component at 200 kHz.

**Figure 10 materials-15-06752-f010:**
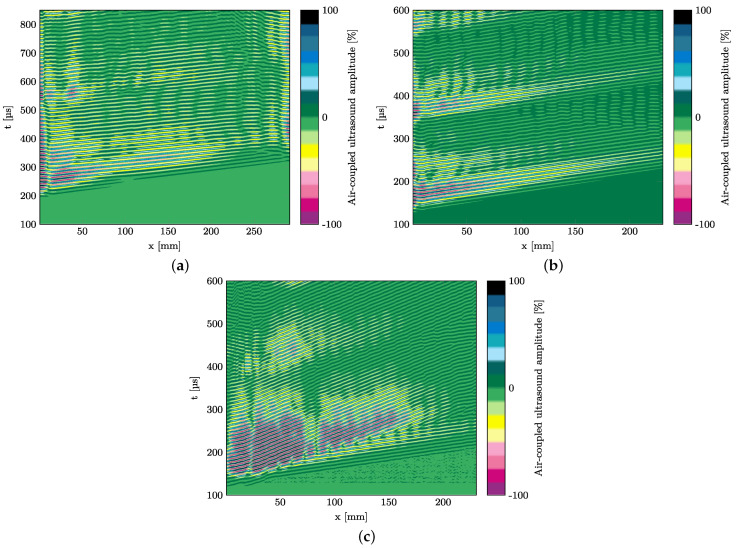
Experimentally determined B—scan of the GFRP strip specimen along the measurement path from actuator to polyimide inlay. Generated using air—coupled ultrasound technique. The polyimide inlay starts at *x* = 100 mm, cf. Figure 2. The measurement with inclined probe shows two modes while the experiment with a perpendicular probe only reveals one mode which is due to the different sensitivities for in—plane and out—of—plane components and the different occurring displacement fields. (**a**) 75 kHz with perpendicular probe. (**b**) 120 kHz with perpendicular probe. (**c**) 120 kHz with probe under an angle of 10°.

**Figure 11 materials-15-06752-f011:**
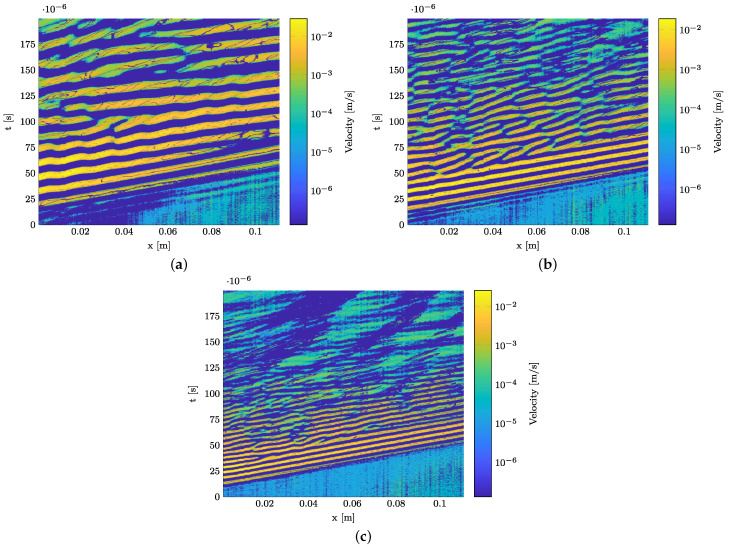
Experimentally determined B-scan of the GFRP strip specimen along the measurement path from actuator to polyimide inlay. The polyimide inlay starts at 0.09 m, cf. Figure 3. In-plane component, generated using 3D LDV measurement. (**a**) 75 kHz. (**b**) 120 kHz. (**c**) 200 kHz.

**Figure 12 materials-15-06752-f012:**
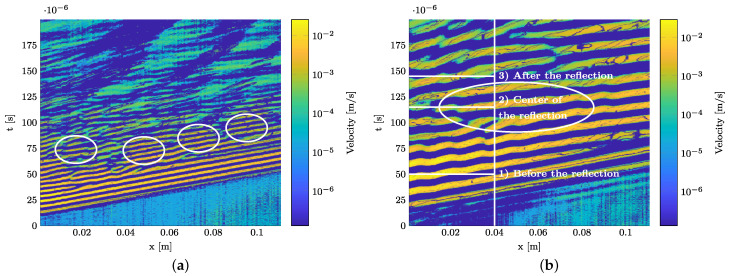
Marked periodic reflections of the $S_{0}$-mode in the B-scan generated using 3D LDV measurement. (**a**) 200 kHz. (**b**) 75 kHz.

**Table 1 materials-15-06752-t001:** Overview of the strip specimen configurations.

Inlay Type	Position
no insert	-
polyimide/no sensor	8th/9th layer (midplane)
polyimide/square	8th/9th layer (midplane)
polyimide/rhombic	8th/9th layer (midplane)
polyimide/flattened rhombic	8th/9th layer (midplane)
polyimide/rhombic	12th/13th layer

**Table 2 materials-15-06752-t002:** Material properties of the individual layers used in the numerical model.

Material	$E_{1}$	$E_{2}$	$G_{12}$	$G_{23}$	$\nu_{12}$	ρ
	[GPa]	[GPa]	[GPa]	[GPa]	[-]	[g/m${}^{3}$]
GFRP DLS1611 (Hexcel Corp.)	54	9.4	5.55	3.75	0.33	1980
Polyimide	2.5	2.5	1.52	1.52	0.34	1420

**Table 3 materials-15-06752-t003:** Phase velocity identification for the fundamental $S_{0}$-mode.

Burst Center Frequency	$c_{\mathrm{phase},\exp,\mathrm{LDV}}\left(S_{0}\right)$	$c_{\mathrm{phase},\mathrm{sim}}\left(S_{0}\right)$
[kHz]	[m/s]	[m/s]
75	2682	2197
120	2500	2193
200	2444	2179

**Table 4 materials-15-06752-t004:** Location of periodic wave reflections at 200 kHz (cf. Figure 12a).

Wave Package	Spatial Starting Point	Temporal Starting Point
	[mm]	[ms]
1	15	0.075
2	40	0.07
3	65	0.08
4	90	0.09

**Table 5 materials-15-06752-t005:** Points in time and location used to identify the first reflected wave packages in Figure 11 with a spatial CWT evaluation, cf. Figure 13.

Burst Center Frequency	Spatial Location		Temporal Point	
		**Before Reflection**	**Center of Reflection**	**After Reflection**
**[kHz]**	**[mm]**	**[μs]**	**[μs]**	**[μs]**
75	40	50	115	145
120	20	50	75	95
200	45	60	70	90

## Data Availability

The raw data of the experiments are available upon request from the authors.

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
