# Peer review of "Influence of a Flat Polyimide Inlay on the Propagation of Guided Ultrasonic Waves in a Narrow GFRP-Specimen"

_materials, 2022, doi:10.3390/ma15196752_

Round 1

Reviewer 1 Report

Reviewers' comments:

Manuscript Number: materials-1925314

Full Title: Influence of a Flat Polyimide Inlay on the Propagation of Guided Ultrasonic Waves in a Narrow GFRP-Specimen.

Comments: 

The manuscript reported on Influence of a Flat Polyimide Inlay on the Propagation of Guided Ultrasonic Waves in a Narrow GFRP-Specimen. The manuscript needs a detailed editing. It cannot be recommended for publication in the present form. I hope the following points would be helpful for the authors.

- Abstract - the authors need to improve with more specific short results and enriched with the brief details of the experimental methodology.

- The introduction section should be improved and the novelty of the work should also be highlighted.

- The experimental section should be detailed especially for the 2.3.2. Measurements of GUW Using the LDV. 

- The authors are obliged to repeat the results part.

- Author should use proper sectional titles throughout the manuscript.

- Requires more information in discussion section as per the application part is concerned.

- More details should add in Evaluation of the Air-Coupled Measurements.

- Author should add a separate section of conclusion. It is very necessary.

- References: make all references in same format for volume number, page numbers and journal name, because it is difficult to searching and reading.

- Some English and grammar mistakes are present that need to be correct to improve the quality of the manuscript.

So that I recommended this manuscript to major revision and for future process.

Reviewer 2 Report

Journal: Materials

Manuscript ID: materials-1925314

Title: Influence of a Flat Polyimide Inlay on the Propagation of Guided Ultrasonic Waves in a Narrow GFRP-Specimen

The authors investigated how integrated polyimide inlays with applied sensor bodies influence the guided ultrasonic wave propagation in narrow glass fiber-reinforced polymer specimens. Preliminary numerical simulations indicate that in a damping-free specimen, the inlays show reflections for the S0-mode propagation. Hence, an air-coupled ultrasonic technique and a 3D laser Doppler vibrometer measurement are used to measure different parts of the propagating waves’ displacement field after burst excitation at different frequencies. No significant reflections on the inlay can be seen in the experiments. However, it is shown that the reflections from the strip specimen’s narrow width cause periodical reflections that superimpose with the excited wave fronts. A continuous wavelet transformation in the time-frequency domain filters discontinuities from the measurement signal and is used for reconstruction of the time signals. The reconstructed signals are used in a spatial continuous wavelet transformation to identify the occurring wavelengths and hence to prove the assumption of reflections from the narrow edges. Since the amplitude of the reflections identified in the numerical data at the polyimide inlays are an order of magnitude smaller than the excited wave packages, it is concluded that material damping of the epoxy resin matrix extincts possible reflections from the inlays.

The paper will be ready for publication after major revision.

The authors need to interpret the meanings of the variables.

Please highlight your contributions in introduction.

What are the main features Figure 1.

The introduction should be supported by recent publications from MDPI such as:

Bistable Morphing Composites for Energy-Harvesting Applications

The abstract should be rewritten to reflect the significance of the proposed work. The current abstract shows a lot of background information.

Conclusion: What are the advantages and disadvantages of this study.

Future work must be included.

Looking and wishes for the revised version.

Round 2

Reviewer 1 Report

The manuscript can published. The authors have answered the questions.

So that I recommended this manuscript accept for publication in Materials.

Reviewer 2 Report

Accept in present form